# Exosomes and Cell Communication: From Tumour-Derived Exosomes and Their Role in Tumour Progression to the Use of Exosomal Cargo for Cancer Treatment

**DOI:** 10.3390/cancers13040822

**Published:** 2021-02-16

**Authors:** Andrea Nicolini, Paola Ferrari, Pier Mario Biava

**Affiliations:** 1Department of Oncology, Transplantations and New Technologies in Medicine, University of Pisa, 56126 Pisa, Italy; 2Unit of Oncology 1, Azienda Ospedaliera Universitaria Pisana, 56126 Pisa, Italy; p.ferrari@ao-pisa.toscana.it; 3Scientific Institute of Research and Care Multimedica, 20099 Milan, Italy; biava.piermario@icloud.com

**Keywords:** exosomes, oncosomes, cancer, exosomal cancer evolution, exosomal cancer therapy, exosomal cancer cells reprogramming

## Abstract

**Simple Summary:**

Recently, within the research community, exosomes, transporters of bioactive molecules involved in many signalling pathways and cell-to-cell communication with the capacity to alter the tumour microenvironment, have been attracting increasing interest among oncologists. These molecules can play multiple roles, e.g., as useful biomarkers in diagnosis, modulators of the immune system, promoters of the formation of the pre-metastatic niches and cancer metastasis and carriers of substances or factors with anticancer properties. This review focuses on the use of exosomes as a novel therapeutic strategy for cancer treatment. Particularly, it highlights the potential of exosomes as carriers of stem cell differentiation stage factors (SCDSFs) for “cell reprogramming” therapy, a promising research field on which we have reported previously. Here, the main characteristics of this treatment and the advantages that can be obtained using mesenchymal stem cell-derived exosomes up-loaded with the SCDSFs as carriers of these factors are also discussed.

**Abstract:**

Exosomes are nano-vesicle-shaped particles secreted by various cells, including cancer cells. Recently, the interest in exosomes among cancer researchers has grown enormously for their many potential roles, and many studies have focused on the bioactive molecules that they export as exosomal cargo. These molecules can function as biomarkers in diagnosis or play a relevant role in modulating the immune system and in promoting apoptosis, cancer development and progression. Others, considering exosomes potentially helpful for cancer treatment, have started to investigate them in experimental therapeutic trials. In this review, first, the biogenesis of exosomes and their main characteristics was briefly described. Then, the capability of tumour-derived exosomes and oncosomes in tumour microenvironments (TMEs) remodelling and pre-metastatic niche formation, as well as their interference with the immune system during cancer development, was examined. Finally, the potential role of exosomes for cancer therapy was discussed. Particularly, in addition, their use as carriers of natural substances and drugs with anticancer properties or carriers of boron neutron capture therapy (BNCT) and anticancer vaccines for immunotherapy, exosomes as biological reprogrammers of cancer cells have gained increased consensus. The principal aspects and the rationale of this intriguing therapeutic proposal are briefly considered.

## 1. Introduction

Exosomes are nano-vesicle-shaped particles secreted under physiological and pathological conditions by various cells, including cancer cells. They export nucleic acids, proteins and lipids that target close or distant cells, thus regulating multiple cellular processes. Cancer research has demonstrated that these particles provide biomarkers for diagnosis [1,2,3], induce apoptosis, modulate the immune system and play a relevant role in promoting cancer development, progression and metastasis. Moreover, specific exosomes are potentially helpful for cancer therapy by various modalities, including re-differentiation of malignant cells. This review first describes the principal morphological-functional characteristics of exosomes. Then, it discusses how tumour-derived exosomes induce cancer evolution and metastasis by affecting the tumour microenvironment (TME) and immune system. Then, the potentially helpful role of exosomes for cancer therapy is considered. Particularly, an increased consensus has been gained by the use of exosomes as biological reprogrammers of cancer cells. Elucidation of the molecular mechanisms of TME reprogramming allows more general conclusions to be drawn for the development of new interventions that are particularly focused on their use as epigenetic reprogramming therapy for malignant cells.

## 2. The Exosomes

The biogenesis of exosomes and their principal characteristics, well described in recent review articles [4,5,6,7], are briefly considered (Figure 1A,B).

### 2.1. The Biogenesis of Exosomes

The term extracellular vesicle (EV) describes any type of membrane particle released by any type of cell into the extracellular space, independent of its biogenesis and composition [8]. The nomenclature and classification guidelines for EVs have recently been updated by the International Society for Extracellular Vesicles ISEV [9]. ISEV endorses EV “as the generic term for particles naturally released from the cells that are cup-shaped and delimited by a lipid bilayer and cannot replicate, i.e., do not contain a functional nucleus” [9]. Endosome-derived EVs are termed exosomes, whereas EVs originating from the plasma membrane are called ectosomes. EVs differing in size, density, sub-cellular origin, function and molecular cargo can be released by a single cell. Thus, heterogeneity is a main feature of EVs and of the consequent exosomes. Due to the difficulty in clarifying the biogenesis of an EV, size is among the suggested different operational terms and the most commonly used among them. Based on their diameter, EVs are divided into “small EVs” (sEVs) and “medium/large EVs” (m/l EVs), with respective size ranges of <100–200 nm (small) and >200 nm (large and/or medium) [9] (Figure 1). Ectosomes are also called microvesicles (MVs), microparticles (MPs), nanoparticles and exosome-like vesicles. Ectosomes include apoptotic bodies (approximately 50–5000 nm in size) and are derived from dying cells by blebbing and fragmentation of cell membranes [10]. Recently, using asymmetric-flow field-flow fractionation (AF4) and at least in part disregarding the size landmarks appointed for EVs by ISEV, two exosome subpopulations have been identified. They have been divided into large exosome vesicles, Exo-L, 90–120 nm and small exosome vesicles, Exo-S, 60–80 nm. Additionally, a population of non-membranous nanoparticles called “exomeres” (approximately 35 nm) has been reported [11]. Exosomes were first described by Pan and Johnstone in 1983 [12]. They originate from the endosomal compartment, also called the endocytic system or endocytic cisterna, a collection of intracellular sorting organelles present in eukaryotic cells. The endocytic system has small membrane curvatures corresponding to different microdomains where cargo, bioactive molecules (proteins and nucleic acids), accumulate. From inward budding of microdomains and their fission form the early endosome (EEs) containing intraluminal vesicles (ILVs), which are the precursors of exosomes. EEs undergo an evolutionary process during which they move to lysosomes where their cargo is degraded, or they merge with the plasma membrane (recycling endosome). Alternatively, exosome precursors, also known as late endosomes/multivesicular bodies (LE/MVB), are released into multivesicular bodies as part of the endocytic machinery [13,14,15] and are then secreted into the surrounding extracellular environment [16,17,18]. The normal biological cellular functions in which exosomes take part govern their number and composition. Exosomes impede enzymatic degradation of their cargo by moving into the blood or extracellular environment [19], while p53 has been reported to have a relevant, although unknown, role in exosome secretion [20]. Neutral sphingomyelinase 2 (nSMase2) is an enzyme limiting ceramide biosynthesis, and its inhibition by ceramides affects exosomal release [21,22,23]. Exosomes are secreted into the surrounding microenvironment and blood as well as in many body fluids (saliva, breast milk, semen, urine, cerebrospinal fluid, amniotic fluid, synovial fluid and sputum) [24,25]. Exosomes and exomeres vary in their lipid composition, and different molecular bioactive molecules have been found through omic analyses. As an example, proteomic analysis showed that exomeres expressed more proteins involved in metabolism, hypoxia, microtubule formation and coagulation compared with large or small exosomal vesicles. Conversely, cargo in large and small exosomes is enriched with mitotic spindle and interleukin (IL)-2/signal transducer and activator of transcription (STAT) 5 signalling, as well as proteins governing endosomal secretion, respectively [11]. It was also found that the differently expressed sialylated glycoproteins (i.e., galectin-3-binding protein) in exosomes and exomeres is organ dependent [11]. Exosomes have been distinguished by density gradient centrifugation into lower (LD-Exo) and higher density exosomes (HD-Exo) [26]. Particularly, up-regulation of solute carrier family 38 member 1 (SLC38A1) has been reported in LD-Exo-treated endothelial cells compared with HD-Exo-treated endothelial cells [26]. It can be inferred that exosome subpopulations have different biological effects on their target cells.

### 2.2. Main Characteristics of Exosomes

Exosomal content depends on the type of donor cell and a regulated sorting mechanism [27]. Different proteins in addition to lipids and nucleic acids (DNA, mRNA and small RNAs such as miRNA, YRNA and tRNA) constitute the exosomal cargo [28,29,30,31]. Some proteins are specific, as they derive from cells and tissues of origin, and others take part in all exosomes [32]. Specific types of exosome proteins are cell-adhesion molecules (CAMs) such as integrins, tetraspanins and MHC class I and II molecules, transferrin receptors (TfR), Rab2, Rab7, flotillin and annexin as well as heat shock (Hsc)70 and 90 proteins. Conversely, cytoskeleton proteins and Alix, which regulates MVB formation, are exosomal proteins taking part in all exosomes [33,34]. Lipids are crucial molecules for exosome shape and biogenesis and for governing homeostasis in the target cells mainly the composition of cholesterol and sphingomyelin [9,35,36]. The inner membrane of MVBs contains lysobisphosphatidic acid (LBPA), a high-density lipid that first allows ILV germination and thereafter, the formation of exosomes [33,37]. Moreover, in ILVs, a hydrolytic phospholipid bis (monoacylglycerol) phosphate/lysobisphosphatidic acid (BMP/LBPA), which is negatively charged, recruits positively charged hydrolases [34,35]. Alix and LBPA interact to induce the budding of MVBs internally [36]. Sphingomyelin and phosphatidylcholine are present similarly in different kinds of microvesicles, although a higher content of sphingomyelin occurs in exosomes. The processes of ubiquitination, plasma membrane myristoylation, prenylation or palmitoylation and the transmembrane glycoprotein CD43 mostly affect the protein content of exosomes [38,39]. Moreover, the exosomal RNA load is tightly regulated by SUMOylated heterogeneous nuclear ribonucleoprotein A2B1 (hnRNPA2B1). This hnRNPA2B1 “binds to miRNAs containing the “shuttling” motif GGAG, therefore resulting in their upload into exosomes” [40]. It has also been found that “AGO2,” a protein joined with the RNA-induced silenced complex (RISC), regulates the loading of miRNAs into exosomes” [41]. Hyper-expressed miRNAs sequences concomitant with hypo-expression of their target mRNAs are loaded into exosomes [42]. Exosomes carry out bioactive molecules expressed on the outer membrane of the EVs or placed in their cargo, to and from the TME. These bioactive molecules account for changes in some principal characteristics of recipient cells and favour cancer growth [27]. Tetraspanins, Endosomal Sorting Complex Required for Transport (ESCRT)-I associated protein, lysosome-associated membrane glycoproteins (LAMP-1 and 2B), Ras-related proteins (Rabs), annexins, MVB-associated protein (Alix-1), HSP60, 70 and 90, adhesion and major histocompatibility molecules (MHC-I and II) are among the most common proteins contained in the exosomes [43]. Genetic material (miRNAs, mRNAs, lncRNAs and DNAs) and lipids also make up part of their cargo [6]. Exosomal transport and the bioactive molecules released to the ECM are governed by Kirsten Rat Sarcoma (RAS)-associated binding (Rab) proteins [19]. A database called ExoCarta involving all data about exosome content is available [44]. Commonly, exosomal autocrine, paracrine and endocrine functions depend on the composition of their cargos. On the other hand, exosome-mediated intercommunication between two different cells [45] can occur by transfer of (1) bioactive material, receptors and proteins to stimulate or suppress signalling pathways, which alter cellular activities and provide specific functions in target cells, respectively, and (2) genetic material to target cells, which acquire new characteristics [46]. Some other findings on the morphology and function of exosomes have recently been reported. In one study [47], secretory carrier membrane protein (SCAMP) 3, which takes part in trans-membrane proteins, has been shown to govern endosomal morphology and composition. Particularly, under certain circumstances, SCAMP 3 concentrated in and was sufficient to recruit hepatocyte growth factor-regulated tyrosine kinase substrate (Hrs) to enlarged endosomes. Endoplasmic reticulum (ER) membrane contact sites (MCSs) were found to correspond to sites where endosomes undergo fission for cargo arrangement. Moreover, the ER membrane protein TMCC1 was concentrated at the ER-endosome MCSs, which are spatially and temporally linked to endosome fission. In another study, it was observed that “when TMCC1 is depleted, endosome morphology is normal and buds still form, but ER-associated bud fission and subsequent cargo sorting to the Golgi are impaired. The endosome-localized actin regulator Coronin 1C was necessary for ER-associated fission of actin-dependent cargo-sorting domains although Coronin 1C was recruited to endosome buds independently of TMCC1” [48]. The authors concluded that the relationship of TMCC1 with Coronin 1C suggests that TMCC1-dependent ER recruitment is strongly governed and that it occurs after cargo is appropriately placed into the bud [48]. The beta-galactoside binding lectin galectin-3 (Gal3) is secreted through a not yet well-defined unconventional mechanism. In a further study [49], Gal3 recruitment and sorting into intra-luminal vesicles and its release were investigated. The endosomal sorting complex required for transport I (ESCRT-I) component Tsg101 and functional Vps4a played a crucial role. Particularly, it was reported that “either Tsg101 knockdown or expression of dominant-negative Vps4aE228Q determined an intracellular Gal3 accumulation at multivesicular body formation sites. In addition, a highly conserved tetrapeptide P(S/T)AP motif in the amino terminus of Gal3 mediating a direct interaction with Tsg101 was identified. Mutation of the P(S/T)AP motif resulted in a loss of interaction and a significant decrease in exosomal Gal3 secretion” [49]. The authors stated that Gal3 is an endogenous non-ESCRT protein P(S/T)AP tagged for exosomal secretion and concluded that this is a unique mechanism that is similar to the direct interaction of syntenin and Alix in exosomal syndecan sorting, which depends on a LYPX(n)L late domain-like motif in syntenin [50]. It can be inferred that many different mechanisms are involved in the transfer of proteins into ILVs [49].

## 3. The Role of Exosomes in Cancer

An emerging concept is that cancer cells can release different subtypes of EVs together with those derived from normal cells [8]. Very large EVs (1–10 μm) from cancer cells, named “large oncosomes” or simply “oncosomes” and derived from the shedding of ILVs, have also been described recently [51]. These large vesicles mainly have aberrant expression of some different oncoproteins, such as myristoylated Akt (MyrAkt)1, heparin-binding epidermal growth factor (HB-EGF) and caveolin-1, suggesting that oncosomes play a key role in cancer cells [52]. Tumour-derived EVs contain many proteinic enzymes involved in glucose, glutamine and amino acid metabolism. This suggests that these EVs play a relevant role in governing the metabolism of cancer cells. [53]. Tumour-derived EVs also provide instructions for extracellular communication, cellular differentiation and migration [54]. The possible release of cell-free nucleic acids (cfNAs) into blood by exosomes has suggested their relationship [55,56], while genetic information transferred from the exosomes to the host cells could transform normal cells by inducing oncogenic mutations and could be involved in promoting genetic instability in target cells [57]. Exosomes contain double-stranded DNA (dsDNA) from the parent cell, and exosomal DNA in tumour-derived exosomes exerts important translational value by acting as a circulating biomarker in the early detection of cancer and metastasis [58]. In addition, lncRNAs are exchanged between gastric cancer cells through exosomes, triggering cancer progression [59]. Nucleic acids in exosomes can be studied by nano-particle tracking analysis (ZetaView), Western blotting, transmission electron microscopy and other techniques [60]. Often, only partial correspondence of tissue profiles with tumour exosome profiles, mainly referring to small RNAs, has been reported [61,62]. The exosomal capability of transferring information involves, in addition to cancer cells, the close and distant TME [63]. During cancer development, oncosomes from brain cancer cells can transfer the oncogenic receptor EGFRvIII to other cancer cells that do not have this receptor [64]. Oncosomes transfer transforming growth factor beta (TGF-β) from cancers to normal fibroblasts, thus promoting myofibroblast differentiation [65]. Conversely, oncosomes from cancer-associated fibroblasts (CAF) inhibit mitochondrial oxidative phosphorylation, thus regulating the metabolism of cancer cells [66]. Exosomes secreted from breast or pancreatic cancers contribute to the formation of metastatic niche carrying telomerase [67] or macrophage migration inhibitory factor (MIF) [68] to the TME. In a study, exosomes carrying TNF-related apoptosis-inducing ligand (TRAIL) re-established apoptosis at the tumour site in vitro and in a preclinical mouse model [69]. The relationships of tumour-derived exosomes with cancer evolution and the immune system are discussed in more detail in the following sections.

### 3.1. Exosomes and the Immune System

There is evidence suggesting the crucial exosomal role of transferring information from cancer cells to immune cells, mainly macrophages, neutrophils, natural killer (NK), dendritic cells and T cells. Immune modulation by cancer cells involves promotion of immune escape and immune tolerance by up-regulation of some genes and proteins [70,71,72]. In the TME, cancer cells induce tumour progression by counteracting cytotoxic T lymphocytes (CTLs) and NK cell activities [73,74]. However, the immunological activities of exosomes in tumours are complex and their concomitant pro- and anti-tumourigenic role in the TME can likely be explained by their functional heterogeneity.

#### 3.1.1. Exosomal Protumour Immune Activities

In pathological conditions such as cancer, a partial block in the differentiation of immature myeloid cells (IMCs) into mature myeloid cells results in an expansion of this population, which has immune-suppressive activity [75]. It has been found that human tumour-released micro-vesicles impair monocyte differentiation into dendritic cells and promote a myeloid cell subset with transforming growth-factor-beta-mediated suppressive activity on T-cell proliferation and their cytolytic functions [76]. Moreover, it has been reported that exosomal HSP72, TLR-2 and MyD88 interacting with immune cells reduce NK and CD4+/CD8+ lymphocytes and impair their cytotoxicities through activation of myeloid-derived suppressor cells (MDSCs) [77]. A recent review [78] mainly reported on the role of tumour-derived exosomes in transferring information from cancer cells to MDSCs. According to this review, TME favours the assembly and delivery of tumour-derived exosomes, which, in turn, promotes tumour progression by inducing the activation and expansion of MDSCs. Expanded MDSCs also increase the immunosuppressive function by producing suppressive molecules. PD-L1, TRAIL, TGF-beta, IL-10, FasL, galectin-9, HSP72 and PGE2 are some of the immunosuppressive molecules that form part of exosomal cargos. Furthermore, they induce the Fas/FasL pathway and apoptosis in CD8+ T cells [79,80,81,82,83,84]. It has also been found that epithelial ovarian cancer-secreted exosomal miR-222-3p can induce the M2 macrophage phenotype in a suppressor of cytokine signalling (SOCS3) STAT3 signal-dependent manner [85]. Macrophages altered by tumour-derived exosomes then can promote tumour progression by inducing angiogenesis and metastasis [86,87,88]. It has been reported that the NKG2D ligand-expressing prostate tumour-derived exosomes selectively induced down-regulation of NKG2D on NK and CD8+ T cells in a dose-dependent manner, leading to impaired cytotoxic function in vitro, thus favouring tumour immune escape [89]. Similarly, exosomes obtained from liquid biopsies of cancer patients impaired the cytotoxicity of NK cells, likely due to an increase in Smad phosphorylation and a decrease in NKG2Dr expression. This supports the notion that tumour-derived exosomes promote cancer growth by inhibiting the immune response [90]. Cancer-associated exosomes carrying miR-24-3p, miR-891a, miR-106a-5p, miR-20a-5p and miR-1908 provoke failure of T-cell activity in nasopharyngeal cancer [91,92]. In pancreatic cancer, immune tolerance in DCs has been found to be related to their derivate exosomes transporting miR-212-3p, which inhibits factor X-associated protein (RFXAP), a regulatory transcription factor for the MHC-II gene [93]. Another in vivo study on pancreatic cancer reported that, in DCs, cancer-derived exosomes down-regulate toll-like receptor 4 (TLR4) and downstream tumour necrosis factor-α (TNF-α) and interleukin-12 (IL-12) cytokines via miR-203, thereby decreasing DC expansion [94]. Increased differentiation and attraction of bone marrow-derived neutrophils are two further modalities by which exosomes can promote cancerogenesis and cancer growth [95]. In a study [96], the transcriptome profiles of primary human normal oral keratinocytes (HNOK) induced by exosomes derived from primary HNOK or head and neck squamous cell carcinoma (HNSCC) cell lines were characterized and compared. In recipient HNOK cells, normal exosomes or tumour-derived exosomes affected many different molecular programmes regarding lipidic metabolism and membrane transport, the anti-inflammatory response, matrix regulation, cytoskeletal remodelling, viral/dsRNA-induced interferon and de-ubiquitination. However, tumour-derived exosomes, unlike normal exosomes, regulated some crucial cellular programs such as the cell cycle, transcription/translation, differentiation, apoptosis and expression of matrix and membrane remodelling. Moreover, CEP55 was suggested to be a potential cancer exosomal marker. The authors stated that “cancer cells may exploit exosomes to confer transcriptome reprogramming that leads to cancer-associated pathologies such as angiogenesis, immune evasion/modulation” [96]. Another recent review [97] focused on the capability of tumour-derived exosomes to induce an immune-suppressive TME favouring disease progression. The importance of miRNAs as modulators of mRNA expression in TME is highlighted. Two recent studies have reported on tumour-cell-derived exosomes or EVs housing PD-L1, which favoured tumour growth by inhibiting T cells. In one of them [98], PD-L1 was transferred from breast cancer cell-derived exosomes to cancer cells with low or absent PD-L1 expression. Then, the functions of T cells were inhibited by the PD-L1 programmed death protein (PD1) interaction. In the other [99], PD-L1 carrying EVs, which were present in the serum or plasma and derived from glioblastoma cells, were directly correlated with the tumour burden.

#### 3.1.2. Exosomal Anti-Tumour Immune Activities

Exosomes can function as a source of tumour antigens that T cells can subsequently recognize. It was demonstrated that exosomes derived from tumours contained antigen-presenting molecules (MHC class-I heat-shock proteins), tetraspanins (CD81) and other tumour antigens (Her2/Neu, Mart1, TRP and gp100) [100], which are very important for stimulation of immune cells against tumours with growth inhibition. The mode of action may involve stimulation of CD4+ and CD8+ T cells. These T cells can cause apoptosis of tumour cells. Moreover tumour-derived exosomes carrying neo-antigens and/or MHC complexes can prime and activate T cells through DCs or directly activate NK cells or macrophages [83,101,102,103,104]. For example, MHC-II, CD80 and CD86 in DCs were up-regulated by exosomes derived from RAB27A-overexpressing cancer cells. This induced CD4+ T-cell proliferation [105]. MART1 tumour antigens were transferred from melanoma-derived exosomes to DCs and thereafter presented to cytotoxic T lymphocytes specific to these antigens [100].

### 3.2. Exosomes in Cancer Development and Progression

Cancer cells over-express EV-associated biogenesis machinery, which allows the release of more exosomes than from normal cells. ESCRT, YKT6, amplifying Rho/ROCK, EGFRvIII, syntenin, heparanase, H-RASV12 and proto-oncogene Src signalling are constituents of this machinery [64,106,107,108,109,110,111,112,113,114]. Hypoxia and a low tumour micro-environmental pH have also been found to favour exosomal traffic, particularly exosomal delivery and uptake in tumour cells [115,116,117,118]. Cancer development and progression is a complex, multistep process driven by many signalling events. In particular, exosomes affect cancer progression by promoting autocrine/paracrine signalling, reprogramming stromal cells, inducing angiogenesis [63] and interfering with the immune system. Oncosomes can transfer oncogenes and their transformed phenotype within subsets of cancer cells. This also accounts for autocrine anti-apoptotic effects of TGF-β1 signalling with the subsequent promotion of cancer proliferation [16] and an increase in growth of the recipient cancer cells [64]. The ZFAS1 lncRNA transferred by circulating exosomes enhances the proliferation of ZFAS1-negative cancer cells [59]. Similarly, some intermediates of the tricarboxylic acid (TCA) cycle are transferred from CAF-derived exosomes to cancer cells, where they contribute to cancer growth through glycolysis modulation and glutamine-dependent reductive carboxylation [66]. A study using an in vitro 3D endothelial-mesenchymal transition (EndMT) model showed that “both tumour-derived exosomes and interstitial fluid flow synergistically promote the EndMT and consequent formation of CAFs through a collagen-based extracellular matrix (ECM) mechanism. The model also demonstrated a homeostatic capability of mesenchymal stem cells (MSC)-derived exosomes, ultimately leading to the recovery of CAFs back to endothelial cells” [119]. Tumour-derived exosomes, by originating endothelial tubule networks [120], favour tumour angiogenesis, which, in turn, provides oxygen and nutrients. The exosome-mediated delivery of miR-9 also promotes differentiation of fibroblasts into CAF with enhanced motility [121]. Moreover, tumour-derived exosomes trigger differentiation of MSC into myofibroblasts with pro-angiogenic and pro-invasive characteristics [122]. Additionally, differentiated MSC secrete growth- and matrix-regulating factors. All this favours tumour growth and progression [122]. Exosomal surface-derived tetraspanin (Tspa) 8 and exosomes carrying miRNA clusters, such as miR-17-92, promote angiogenesis by up-regulating angiogenesis-related genes [17] and stimulating endothelial migration and tube formation [123]. Neo-angiogenesis also can be due to hyper-expression of vascular cell adhesion molecule-1 (VCAM-1) and intercellular adhesion molecule-1 (ICAM-1) largely found in endothelial cells treated with exosomes released from chronic myelogenous leukaemia (CML) cells [124]. A capacity to increase motility and to induce tube network development has been observed in endothelial cells reprogrammed by these exosomes [124]; miR21 governs PTEN/PI3K/AKT signalling and inhibits apoptosis in gastric cancer [125]. Tumourigenesis and cancer progression are also favoured by exosome-dependent mechanisms, as they increase the differentiation of bone marrow neutrophils and recruit neutrophils to cancer cells [94]. Heparan sulphate proteoglycans (HSPGs) also reside on the surface of EVs. A recent review article [126] discussed, mainly focusing on tumour progression, multiple functions that allow HSPG to interfere with the capability of EVs to transfer information in different contexts. In particular, EV-associated PGs are proposed “as dynamic reservoirs and chaperones of signalling molecules with potential implications in ligand exchange between EVs and tumour target cells” [126]. In non-small-cell lung cancer (NSCLC), long non-coding RNAs (LncRNAs) have been reported to significantly affect cancer evolution. In a study [127], the impact of lncRNA UFC1 in NSCLC has been investigated. Particularly, the interaction between UFC1 and enhancer of zeste homolog 2 (EZH2) and the binding of EZH2 to the PTEN gene promoter were assessed. In tumour tissues, serum and serum exosomes of NSCLC patients, UFC1 was overexpressed concomitant with tumour infiltration. Following UFC1 knockdown, induction of cell cycle arrest and apoptosis and inhibition of NSCLC progression occurred. Mechanistically, it was uncovered that UFC1 binds to EZH2 and mediates its increase at the promoter region of PTEN gene. This accounts for trimethylation of H3K27 and the inhibition of PTEN expression via EZH2-mediated epigenetic silencing.

### 3.3. Exosomes and Cancer Metastasis

Soluble or vesicle-enclosed bioactive molecules derived from cancer cells play a relevant role in remodelling the architecture of the extracellular matrix and in reprogramming cells of the TME in distant organs to prepare suitable pre-metastatic niches. Hart et al., 1980, consistent with the Paget’s “seed and soil” hypothesis, proposed the “non-random pattern of cancer metastasis” [128]. Nonetheless, so far, the mechanisms governing organ-specific metastasis are poorly understood. Tumour-derived exosomes contribute to prime pre-metastatic niches by inducing the function of hematopoietic progenitors derived from bone marrow and expressing the vascular endothelial growth factor receptor 1 (VEGFR1) through the transfer of exosomal MET oncoprotein [129]. It has been described that tumour-derived exosomes favour the establishment of pre-metastatic niches by inducing the activation of Src, and up-regulation of pro-inflammatory S100 genes in the involved cells of target organs. Similarly, TGF hyper-expression of Kupffer cells in the liver by MIF-1 carried out by pancreatic tumour exosomes has been found. Subsequently, it has been observed that stellate cells produce an enhanced amount of fibronectin [68]. The remodelled TME further increases the arrival of macrophages from bone marrow, thus favouring the arising of a pre-metastatic niche in the liver [68]. In a study [130], a positive correlation of exosomal integrins α6β4 and α6β1 with lung metastasis, and of exosomal integrin αvβ5 with liver metastasis, was observed. Based on these findings, a metastatic organotropism was hypothesized. Namely, it was proposed that the integrin expression profiles of circulating plasma exosomes act as “ZIP codes” addressing exosomes to well-defined tissues/organs [130]. Various studies have also reported that tumour-derived exosomes release extracellular matrix remodelling enzymes such as matrix metalloproteinase (MMP)2 or MMP9, which promote cancer invasion and metastasis through extracellular matrix degradation [131]. Invadopodia are actin-rich protrusions of the plasma membrane of cancer cells that join with degradation of the ECM in cancer invasion and metastasis [132,133]. Invadopodia biogenesis and exosome secretion are often two concomitant processes needed for exosome-mediated cancer diffusion [134,135]. Rab27a knockdown significantly decreased the development of mature invadopodia, as well as extracellular matrix digestion through exosomal biogenesis or secretion inhibition [135]. Considering that invadopodia play a crucial role in exosomal secretion, this suggests that exosomes govern invasiveness in a synergistic way [135]. Moreover, it has been reported that the metastatic potential of cancer cells can be exerted by tumour-derived exosomes through the transfer of miRNAs and the subsequent alterations in gene expression and promotion of mesenchymal to-epithelial transition (EMT). Particularly, in experimental models, miR-200-expressing tumours and extracellular vesicles from these tumours induced metastasis of otherwise weakly metastatic cells, and these cells were capable of colonizing distant tissues in an miR-200-dependent manner [136]. An investigation [137] reported that hepatocarcinogenesis is strongly affected by deregulated exosomal miRNAs. In this study, transforming growth factor beta (TGF-beta) was used to induce EMT in a cellular line of hepatic cancer (Hep3B). Subsequently, exosomes derived from unstimulated Hep3B cells (Hep3B exo) or TGF-β-stimulated EMT Hep3B cells (EMT-Hep3B exo) were analysed. Higher expression of the exosomal marker proteins CD63 and TSG101 was found in both of them, with the highest levels occurring in the latter. EMT-Hep3B exo stimulated Hep3B cell proliferation while inducing migration and invasion of Hep3B and 7721 cells. Within exosomes, 119 up-regulated and 186 down-regulated miRNAs and 156 up-regulated and 166 down-regulated mRNA sequences were found in the EMT-Hep3B exo in comparison with the control Hep3B exo. Gene ontology (GO) analysis showed a correspondence between differentially expressed miRNAs and differentially expressed mRNAs; the latter were mainly involved in metabolic pathways and pathways in cancer. A regulation of the expression of GADD45A mRNA and protein by miR-374a-5p was also hypothesized. Inhibition of miR-374a-5p in Hep3B cells resulted in exosomal inhibition of proliferation and invasiveness in HCC cells. Extracellular galactin-3 interacting with other cells in the TME contributes to metastasis [138]. The blood–brain barrier (BBB) can be structurally and functionally interrupted by exosomes derived from metastatic cancer cells [139,140]. In particular, it has been found that cancer-secreted miR-105 or EVs, released by brain metastatic cells and containing miR-181c, promote brain metastasis through destruction of vascular endothelial barriers. These barriers are impaired addressing tight junction proteins or inducing abnormalities of the cytoskeleton [139,140]. Frizzled (FZD) is a family of G protein-coupled receptor proteins that act as soluble mediators of the Wnt signalling pathway and plays a crucial role in embryonic development and carcinogenesis of the gastroenteric tract. In a study [141], exosomal FZD10 protein and FZD10-mRNA were obtained from the culture medium of cellular lines of untreated colorectal, gastric and hepatic cancers and cholangiocarcinoma. A significant decrease in the FZD10 protein and FZD10-mRNA levels contemporaneously with a strong reduction in their viability occurred in FZD10-mRNA-silenced cells and in their exosomes compared with the controls. Accordingly, silenced cells incubated with exosomes derived from culture medium from the same untreated cells recovered viability as well as the FZD10 and FZD10-mRNA levels. This suggests that exosomal FZD10 protein and FZD10-mRNA may induce tumour growth and metastasis. The role that some bioactive molecules play in cancer evolution as exosomal cargo is shown in Table 1.

## 4. Use of Exosomes in Cancer Therapy

Having described the role that tumour-derived exosomes play as carriers of specific substances that can spread tumours in an organism, we would now like to illustrate in the next paragraphs the use of exosomes as carriers of many substances with anticancer properties.

### 4.1. Exosomes as Carriers of Natural Substances or Chemotherapeutical Drugs with Anticancer Properties

Exosomes can function as drug delivery carriers in cancer therapy (Table 2) [146]. Taking into account that exosomes are very small particles, they can penetrate the cell membrane easily. Many studies about exosomes as drug delivery systems have been conducted, and their high specificity in targeting cancer cells has been widely reported [147]. In an experimental study, direct and specific targeting of oncogenic KRAS by engineered exosomes from normal fibroblast-like mesenchymal cells was described [148]. In 2016, researchers used exosomes into which they introduced curcumin to treat cancer patients [149]. Curcumin is a known biological substance with recognized anticancer properties. Their studies proved that exosomes can transfer curcumin into H1299 cells, causing TCF21 up-regulation with anticancer effects [149], while curcumin alone is inefficient because it lacks bioavailability [150,151]. In addition, some studies have been conducted on the ability of exosomes to transport chemotherapeutic agents. For example, in 2014, some researchers tested the capacity of exosomes as a drug delivery system to transfer chemotherapeutic agents such as doxorubicin [152,153]. The results showed that when administered in exosomes, doxorubicin inhibited cell proliferation in BALB/c nude mice. Exosomes were obtained by mouse immature dendritic cells (imDCs) to significantly decrease the side effects and immunogenicity. An exosomal membrane protein (Lamp2b) fused to αv integrin-specific iRGD peptide (CRGDKGPDC) was incorporated into imDCs by engineering to favour tumour targeting [154]. Electroporation allowed loading of these purified exosomes with doxorubicin, with an encapsulation efficiency of up to 20%. These exosomes, after intravenous injection, greatly inhibited tumour growth, and this treatment was well tolerated. In fact, intravenously injected targeted exosomes showed efficient doxorubicin delivery leading to greater inhibition of various tumour cell lines than doxorubicin alone. In addition, low toxicity was demonstrated when doxorubicin was transferred using exosomes in comparison with the use of doxorubicin alone. In another experimental investigation [155], a potent chemotherapeutic agent, paclitaxel (PTX), to treat MDR cancer was evaluated. Different methods of loading exosomes released by macrophages with PTX (exoPTX) that showed high loading efficiency and drug release were developed. Interestingly, incorporation of PTX into exosomes enhanced cytotoxicity more than 50 times in drug-resistant MDCKMDR1 (Pgp+) cells. Moreover, other studies demonstrated a relevant co-localization of airway-delivered exosomes with cancer cells and a strong anticancer activity in a model of murine Lewis lung carcinoma pulmonary metastases. Authors concluded that exoPTX can deliver various chemotherapeutics to treat drug-resistant cancers. It is possible to conclude that exosomes possess the capacity to act as a delivery system of chemotherapeutic agents and that they can increase therapeutic effects and reduce side effects in comparison with the use of the anticancer drug alone.

### 4.2. Exosomes as Carriers of Boron Neutron Capture Therapy (BNCT)

BNCT joins selective accumulation of ^10^B carriers in tumour tissue with neutron irradiation. So far, more often, BNCT has been investigated in experimental clinical studies for recurrent malignant gliomas, head and neck cancers as well as multiple lung metastases using neutron beams derived from research reactors [158,159]. Phenylboronic acid-functionalized nanocarriers for specific targeting to sialic acid groups over-expressed on tumour cells appear to be a very promising mechanism [160]. Currently, two low-molecular-weight boron-containing drugs are being used clinically, boronophenylalanine (BPA) and sodium borocaptate (BSH). However, in the last years, because of some limitations, great effort has been made to develop nanoscaled boron-containing delivery agents with more favourable biodistribution and uptake for clinical use. These potential nanocarriers include exosomes in addition to other nanoparticles of various types [161].

### 4.3. Exosomes as Anticancer Vaccines for Immunotherapy

In recent years, exosomes also have been experimentally employed as a cancer immunotherapy treatment [156,157]. Particularly, they have been proposed as anticancer vaccines due to their capacity to carry antigens and MHC-peptide complexes and to promote helper T-cell immune responses. In a study [162] conducted in a B16 melanoma model, vesicles released as exosomes or microvesicles (MV) and alternatively as apoptotic blebs or vesicles (ApoV) from the plasma membrane of apoptotic cancer cells were characterized. Distinct types of surface and cytoplasmic molecules (tetraspanins, integrins, heat shock proteins and histones) were expressed in the different vesicle types. Interestingly, mice immunized with antigen-pulsed ApoV and challenged with B16 tumours showed significantly longer tumour-free survival than mice immunized with ovalbumin-pulsed exosome vaccine or ovalbumin-pulsed microvesicle vaccine. In a phase I clinical trial, ascites-derived exosomes (Aex) in combination with the granulocyte-macrophage colony-stimulating factor (GM-CSF) were used in the immunotherapy of colorectal cancer (CRC). Overall, 40 patients with advanced CRC were recruited to the study and randomly assigned to treatments with Aex alone or Aex plus GM-CSF. Both therapies were safe and well tolerated; however, Aex plus GM-CSF but not Aex alone induced a tumour-specific antitumour cytotoxic T-lymphocyte (CTL) response [163].

### 4.4. Exosomes as Biological Reprogrammers for Cancer Treatment

The composition of exosomes has been clarified in the last years and, as previously reported, many different proteins have been identified inside exosomes. They carry specific nucleic acids, such as mRNA, that have been shown to induce phenotypic changes in recipient cells. Exosomes from different stem cells and from stem cells at different developmental stages may carry specific information. Exosomes contain proteins and nucleic acids that directly reflect the metabolic state of the cells from which they originate. Exosomes can regulate multiple genes in an epigenetic manner and are capable of producing multiple signal transduction cascades. Communication depends on specific crosstalk between exosomes and the target cells. This paracrine mechanism can mediate cell-to-cell communication via direct receptor stimulation of target cells and the horizontal transfer of genetic material to cells prone to avidly receive this information. Using exosomes for cancer therapy has many potential benefits [164]. It was also demonstrated that exosomes secreted by mesenchymal stem cells (MSC) have the ability to control tumour growth due to the specific substances they transfer. Exosomes derived from MSC contain multiple cargoes and proteins that may control different relevant metabolic pathways of malignant cells [165]. In 2012, researchers evaluated if exosomes from MSC of human bone marrow can inhibit in vivo and in vitro growth of multiple tumours. The study revealed that exosomes secreted from MSC of human bone marrow inhibited cell cycle progression and induced apoptosis in different cancer cells such as hepatocellular carcinoma, ovarian tumour and Kaposi’s sarcoma [166]. Another series of studies emerging in cancer therapy regard the possibility of cancer cell reprogramming. The term “reprogramming” was initially introduced to identify the transformation of a normal adult somatic cell into an embryonic-like stem cell, so-called induced pluripotent stem cells (iPS). The issue of cell reprogramming has now been extended to cancer (stem) cells to define any genetic or epigenetic intervention aimed at inducing differentiation of these cells into a normal phenotype and/or forcing them to become terminally differentiating cells [167]. In order to differentiate, tumour cells must go through cell regulation pathways. These pathways can be controlled by multiple cell differentiation factors. For example, exosomes derived from mesenchymal stem cells can function as differentiating factors, because they transfer particular instructions to direct cancer cells toward differentiation or apoptosis. Stem cells are found in and can be isolated from multiple tissues of our bodies, such as adipose tissue, bone marrow and the umbilical cord. Some functions of these cells are to replace damaged cells and to cause differentiation. Very interesting are studies on umbilical cord stem cells (hUCMSCs) derived from Wharton’s jelly due to multiple advantages they possess: in fact, hUCMSCs derived from Warton’s jelly possess a unique transcriptome that shows pro-apoptotic and anticancer properties [168]. A study conducted in 2012 demonstrated that human Wharton’s jelly stem cell (hWJSC) extracts significantly attenuated tumour growth in three different cancer cell lines in vitro [169] and in another study, it was observed that exosomes from umbilical cord-derived MSC inhibited the growth of breast cancer cells in vitro and in vivo [170].

### 4.5. Stem Cells Differentiation Stage Factors (SCDSFs): Other Reprogramming Factors of Cancers, besides Exosomes

Another interesting path of studies about cancer cell reprogramming regards the research on stem cell differentiation factors and their role in reprogramming cancer (stem) cells. These studies originated on the basis of many previous studies, which demonstrated that cancer development can be prevented during embryonic life due to the factors present during organogenesis, as recorded in previous reviews [171]. In fact, studies conducted in our laboratory demonstrated that not only many different human tumour cell lines (glioblastoma multiforme, hepatocellular carcinoma, melanoma, breast cancer and kidney adenocarcinoma) but also non-solid tumours, such as acute lymphoblastic leukaemia, treated with factors taken from zebrafish embryos during organogenesis demonstrated a significant slowdown in their growth only when treated with factors taken at the precise moments of cell differentiation stages [172]. The effects of SCDSFs on tumour growth were also observed in vivo after subcutaneous injection of primary Lewis Lung Carcinoma cells into C57BL/6 female syngeneic mice weighing 18–20 g. A highly significant difference was noted (*p* < 0.001) between treated and control mice in favour of the treated mice [171,173]. All these experiments confirm the hypothesis that only when the factors taken during the stages of differentiation are used, it is possible to direct tumour cells towards the normal path of differentiation. These factors appear in the phases of cell differentiation, and they are absent in the stages of mere multiplication. It has been demonstrated that SCDSFs taken from zebrafish embryos contain apoptosis-inducing proteins that can act on colon cancer cells [174]. In another study, all types of proteins taken from zebrafish embryos after the beginning of stem cell differentiation were identified using liquid chromatography–mass spectrometry (LC-MS/MS) analysis, after the in-gel digestion procedure [173]. The identified proteins, which represent 98% of the molecules isolated from SCDSFs (the remaining 2% of the molecules are represented by nucleic acids), included multiple forms of yolk vitellogenin, heat shock protein (e.g., HSP8 and HSP70), which are important for the immune response, and other proteins able to regulate mitochondrial metabolism, etc. These proteins are implicated in many pathways such as signalling in cell cycle regulation, protein trafficking, chaperoning and protein synthesis and degradation. It was confirmed that these proteins have a reprogramming or apoptotic effect on cancer cells because they act to regulate transcriptional activation of p53 [175] or a post-translational modification of the protein of the retinoblastoma (pRb), able to reprogram cancer cells [176]. Moreover, apoptotic events as well as cell differentiation events were studied, in order to understand the consequences of cell cycle regulation in tumour cells induced by differentiation factors. The analysis was carried out on colon adenocarcinoma cells, showing activation of an apoptotic pathway dependent on p73, as well as an increase in the cell differentiation marker e-cadherin [174]. In addition, in order to ascertain if these embryonic factors could synergistically/additively interact with 5-fluorouracil (5-Fu), whole cell-count, flow-cytometry analysis and apoptotic parameters were recorded in human colon cancer cells (Caco2) treated with zebrafish stem cell differentiation stage factors (SCDSF 3 µg/mL) in association or not with 5-Fu in the sub-pharmacological therapeutic range (0.01 mg/mL). Cell proliferation was significantly reduced by SCDSF; meanwhile, SCDSF+5-Fu leads to an almost complete growth inhibition. SCDSF produces a significant apoptotic effect; meanwhile, the association with 5-FU leads to an enhanced additive apoptotic rate at both 24 and 72 h. SCDSF alone and in association with 5-Fu triggers both the extrinsic and the intrinsic apoptotic pathways, activating caspase-8, -3 and -SCDSF and 5-Fu alone exerted opposite effects on Bax and Bcl-xL proteins; meanwhile, SCDSF+5-Fu induced an almost complete suppression of Bcl-xL release and a dramatic increase in the Bax/Bcl-xL ratio [177]. These data suggest that zebrafish embryo factors could improve chemotherapy efficacy by reducing anti-apoptotic proteins involved in drug-resistance processes. This information is congruent with other studies, which demonstrate that differentiation factors can possess epigenetic regulators that are able to regulate cancer cells by activating new pathways. In particular in a recent study, it was observed that SCDSFs significantly antagonize proliferation of breast cancer cells, because they not only reduce cell proliferation and enhance apoptosis but also dramatically inhibit both invasiveness and the migrating capabilities of cancer cells, controlling, in this way, tumour spread and metastasis. Moreover, it was demonstrated that SCDSFs are also able to inhibit migration and invasiveness of breast cells in the epithelial-mesenchymal transition phase following TGF-β1 stimulation. The reversion program involves modulation of the E-cadherin/β-catenin pathway, cytoskeleton remodelling with dramatic reduction in vinculin, as well as down-regulation of translationally controlled tumour protein (TCTP) and the concomitant increase in p53 levels [178]. In addition, it is noteworthy that SCDSFs have a very important role in cancer treatment when they are transferred by exosomes. In fact, it was demonstrated that SCDSFs contained in exosomes derived from mesenchymal stem cells express functional respiratory complexes, which may promote in cancer cells aerobic metabolism able to counteract the progression of cancer cells that use anaerobic metabolism (Warburg effect) for their progression [179,180,181,182,183,184,185,186].

### 4.6. Clinical Trials on Patients Treated with SCDSFs

#### 4.6.1. Hepatocellular Carcinoma in the Intermediate–Advanced Stage

A randomized controlled trial was conducted in 179 patients with an intermediate-advanced stage of hepatocellular carcinoma for which no treatments of consolidated efficacy were possible. On the basis of the previous studies described above, it was possible to develop a nutraceutical product containing the substances collected in the stages of cell differentiation in which they had demonstrated great efficacy in controlling tumour growth. Considering the low molecular weight of these proteins, sublingual absorption was suggested. The results showed a statistically significant difference in favour of the treated patient group, in comparison with the non-treated patient group. In all, 19.8% of the patients demonstrated a regression and 16% of the patients demonstrated a stabilization, with an overall survival after 40 months of more than 60% of the patients who responded, compared with 10% of the non-responding patients. An improvement of performance status was registered in 82.6% of the patients [187]. A more recent clinical trial conducted by the Scientific Institute of Research and Care Humanitas of Milan on patients with hepatocellular carcinoma in an advanced stage has confirmed the role of SCDSFs in producing a complete response [188].

#### 4.6.2. Colon Cancer in an Advanced Stage

Lastly, a recent clinical trial conducted by the Institute of Oncology of University La Sapienza of Rome in a group of patients with advanced-stage colon cancer treated with Regorafenib alone, in comparison with another group treated with Regorafenib plus SCDSFs, demonstrated a statistically significant increase in survival in the latter group [189]. Other papers about the efficacy of SCDSFs in cancer treatments were published by different authors who suggested the use of SCDSFs as integrative treatment to the traditional therapies of consolidated efficacy [167,190,191]. It is also worth noting that a declaration of a committee of oncologists, published in a recent book, suggests the use of SCDSFs as integrative treatments in oncology [192].

## 5. Discussion

Based on the discussion thus far, it is not difficult to propose the hypothesis that cancer cells may revert to a normal or less malignant phenotype through MSC-derived exosomes up-loaded with SCDSFs factors. Theoretically, relevant advantages could be derived from this approach to cancer treatment. MSC-derived exosomes themselves have been demonstrated to transfer genetic information with efficacious differentiation capability. Enriching their cargos with further SCDSFs taken from zebrafish embryos should be a relatively easy technical achievement. The increased number of differentiating factors in their cargo should substantially enhance the reversion of pathological molecular pathways to a more normal phenotype. This procedure, which occurs physiologically and with high specificity in targeting malignant cells, should permit the joining of high therapeutic effectiveness with low or no side effects. Moreover, it is not for us to disregard the possibility of combining this biological approach with more conventional therapies. However, it is clear that we are only at the beginning of this route and that to follow through, well-designed prospective randomized clinical trials are needed.

## 6. Conclusions

We can conclude that exosomes are very important nanoparticles for intercellular communication. Due to their ability to pass bio-information to other cells, they can perform multiple functions in the tissues. Cancer treatment needs to be improved by increasing effectiveness and specificity. Research has proven that exosomes can be very specific. They carry multiple cargo types, proteins, microRNA, mRNA and nucleic acids that can have specific actions in target cells. Therefore, up-loading selected cargos could allow the provision of specific treatments for different cancer types. The therapeutic ability of exosomes to fight cancer can be very promising. Exosomes are relatively easy to use, specific and non-toxic. These characteristics have great potential for implementation of exosomes as a therapy for cancer. Cancer cells can revert to a normal phenotype in the presence of SCDSFs, which are able to reprogram cancer cells, as demonstrated in the reviewed studies. This epigenetic regulation of cancer cell DNA may be accomplished by modifying DNA methylation and histone modification in the promoter region of the silenced tumour suppressor genes, generating all the reprogramming mechanisms here described. We are now conducting experiences introducing SCDSFs into exosomes, and we are verifying if we can obtain better results both at experimental and clinical level. These researches are ongoing and we will illustrate the results obtained from these researches as soon as they are completed. However, more research needs to be done to fully understand, specifically select and effectively apply these differentiation-promoting exosomes. We need to better understand the physiological effect of different exosomes of diverse developmental stages and those arising from cells of different tissues. It is critical to know more about the composition of these nanovesicles so that we can understand better how to use them in cancer treatment.

## Figures and Tables

**Figure 1 cancers-13-00822-f001:**
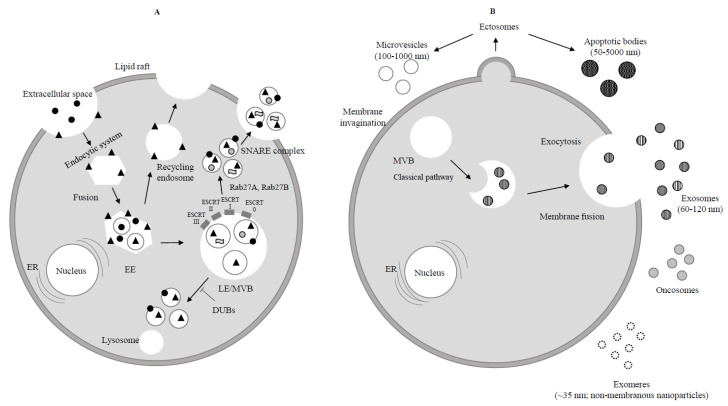
Biogenesis of exosomes. (**A**) Endocytosis begins at the cytoplasmic membrane. Early endosomes (EEs) form by the fusion of uncoated endocytic vesicles. EEs either return to the plasma membrane (recycling endosome) or convert into late endosomes/multivesicular bodies (LEs/MVBs). Protein sorting of ILVs (intraluminal vesicles) occurs through endosomal sorting complexes required for transport (ESCRT) dependent or independent mechanism. ESCRT 0, ESCRT I, ESCRT II, ESCRT III are four components of ESCRT machinery that ubiquitinate the substrates on the part of the inward budding endosomal membrane. ILVs are ready to be degraded by lysosome or rescued by de-ubiquitinating enzymes (DUBs) through Rab GTPases (Rab27A and Rab27B) which allow the MVBs to move to the cell periphery. Finally MVBs through the SNARE complex fuse with the plasma membrane and lead to exocytosis (release of ILVs as exosomes to the extracellular space). 
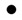
 Extracellular protein; 
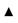
 Receptor; 
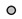
 Lipid; 
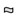
 RNA. (**B**) Ectosomes derive from the plasma membrane by direct gemmation. Large oncosomes, as microvesicles originate from plasma membrane [8]. Exosomes are derived and secreted through a “classical” pathway. ER = endoplasmic reticulum. 
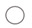
 Microvesicle; 
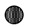
 Apoptotic bodies; 
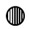
 Exosome; 
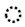
 Exomere; 
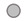
 Oncosome.

**Table 1 cancers-13-00822-t001:** Role that some bioactive molecules play in cancer evolution as exosomal cargo.

Exosomal Cargo			Biological Function	Cancer Type	Ref.
Bioactive Molecule	Type	Mechanism			
Delta-like	Protein	Notch signal inhibition	Increased angiogenesis	Tumour xenograft model	[142]
EGFR VIII	Protein	Promotes Akt and MAPK pathways	Increased anchorage-independent growth	Glioma	[64]
Integrins	Protein	Induce src and up-regulate proinflammatory s-100 genes	Leading exosomes to specific tissues	Breast	[130]
MET	Protein	Promotes MET signal	Increased prometastatic activity of bone marrow cells	Melanoma	[129]
MIF	Protein	Promotes TGF-beta signal-induced fibronectin production	Favoured premetastatic niche formation at liver	Pancreatic	[68]
TGF-beta	Protein	Promotes SMAD-related signal Induces mesenchymal stem cell differentiation into myofibroblasts	Induced fibroblast FGF2 production, increased cancer proliferation and invasiveness	Prostate	[65,122]
CD63, CD81, HSP90, HSP70, TNF1-alpha, MMP2, MMP9, annexin-II	Proteins	Increase TGF-beta 2, TNF1-alpha, IL-6, TSG101, Akt, ILK1, beta-catenin signalling; remodelling of epithelial adherent junctions	Increased stemness, metastasis and CAFs formation	Prostate	[143]
Fasl, TGF-beta, galectin-9, HSP72	Proteins	Transfer of Fasl, TGF-beta, NKG2D ligands, galectin-9, HSP72 into immune cells	Evasion of immune responses	-	[79,80,81,82,83,84]
Integrin av-beta3	Protein	Promotes cell migration on its ligand, vitronectin	Promotion of a migratory phenotype	Prostate	[144]
PD-L1	Protein	PD-L1 up-regulation	Immune checkpoint regulator	Melanoma	[145]
Tspan8	Protein	Selective recruitment of proteins and mRNA	Induces endothelial cell proliferation, migration and sprouting	Rat adenocarcinoma model	[17]
miR-9	miRNA	Modulation of genes involved in cell motility and extracellular matrix remodelling pathways	CAF-like induction	Breast	[121]
miR-21	miRNA	Regulates PTEN/PI3K/AKT signal	Apoptosis inhibition	Gastric	[125]
miR-105	miRNA	Down-regulates tight junctions	Vascular endothelial barrier destruction	Breast	[139]
miR-181c	miRNA	Down-regulates PDPK1/cofilin signal	Blood-brain barrier destruction	Breast	[140]
miR-200	miRNA	Regulates gene expression and EMT	Induction of cancer metastatization to the lung	Breast	[136]
miR-222-3p	miRNA	Regulates SOCS3/STAT3 pathway	Induction of TAM polarization	Epithelial ovarian	[85]
ZFAS1	lncRNA	Governs MAPK signal and EMT transcription factors	Induction of cell cycle progression and EMT	Gastric	[59]
hTERT mRNA	mRNA	Converts nonmalignant fibroblast into telomerase positive cells	Phenotypic changes (increased proliferation and extension of life span)	Pancreatic and lung	[67]
TCA-cycle intermediate	Metabolite	Governs mitochondrial oxidative phosphorylation, glycolysis, glutamine-dependent reductive carboxylation	Down-regulation of mitochondrial function and up-regulation of glucose metabolism	Prostate	[66]

EGFR: epidermal growth factor; Akt: protein kinase B; MAPK: mitogen-activated protein kinase; MIF: migration inhibitor factor; TGF: tumour growth factor; FGF: fibroblast growth factor; HSP: heat shock protein; TNF: tumour necrosis factor; MMP: matrix metalloproteinase; IL: interleukin; ILK: integrin-linked kinase; CAFs: cancer-associated fibroblasts; Fasl: Fas ligand; NKG2D: natural killer group 2D; PD-L1: programmed death-ligand 1; Tspan8: tetraspanin-8; miR: micro RNA; PTEN: phosphatase and tensin homolog; PI3K: phosphoinositide 3-kinase; PDPK1: 3-phosphoinositide-dependent protein kinase 1; EMT: epithelial to mesenchymal transition; SOCs: suppressor of cytokine signalling proteins; STAT: signal transducer and activator of transcription 3; TAMs: tumour associated macrophages; hTERT: human telomerase reverse transcriptase; TCA cycle: tricarboxylic acid cycle (Krebs’ cycle).

**Table 2 cancers-13-00822-t002:** Some in vivo experimental studies and clinical trials for exosomal cancer therapy.

Exosomal Therapy	Type of Study	Mechanism	Cancer	Ref.
Exosome delivery of anticancer drugs across BBB	In vivo (zebrafish model)	Cytotoxic effects	Brain	[147]
MSC-derived exosomes with KRAS G12D siRNA	Clinical trial	KRAS G12D signalling and cancer cell growth inhibition	Pancreatic with KRAS G12D mutation	[148]
Exosomes derived from curcumin-pretreated H1299 cells	Nude mice	TCF21 up-regulation	Lung cancer	[149]
imDCs exosomes fused to av integrin-specific IRG peptide loaded with DOX	In vivo (BALB/c nude mice)	Cytotoxic effects	Breast	[154]
Exo PTX	C57BL/6 mice injected with LL-M27 cells	Cytotoxic effects	Lewis lung carcinoma pulmonary metastases	[155]
DEX loaded with MAGE cancer antigens	Clinical trial	Immunotherapy	NSCLC	[156]
IFN-gamma dendritic cell-derived exosomes loaded with MHC class I- and class II-restricted cancer antigens	Clinical trial	Maintenance immunotherapy	NSCLC	[157]

BBB: brain–blood barrier; imDCs: mouse immature dendritic cells; Exo PTX: exosomes derived from macrophages and loaded with paclitaxel; DOX: doxorubicin; NSCLC: non-small cell lung cancer; MSC: mesenchymal stem cells; DEX: dendritic cell-derived exosomes; KRAS: Kirsten Rat Sarcoma; IFN: interferon; MHC: major histocompatibility complex.

## Data Availability

No new data were created or analyzed in this study. Data sharing is not applicable to this article.

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
