# Peer review of "Exosomes and Cell Communication: From Tumour-Derived Exosomes and Their Role in Tumour Progression to the Use of Exosomal Cargo for Cancer Treatment"

_cancers, 2021, doi:10.3390/cancers13040822_

Round 1
Reviewer 1 Report
no other comments
Author Response
Thank you
Reviewer 2 Report
Authors have made necessary changes as requested.
Author Response
Thank you
Reviewer 3 Report
The authors have fully addressed my concerns and taken into account my comments. The manuscript is now better organised and easier for the readers to digest. It will be very interesting to follow the progress in this field. Well done.
Author Response
Thank you
Reviewer 4 Report
The review proposed by Nicoli et al. was improved. However, there are still some problems with several parts and most of the titles does not reflect their content. In part 3, since oncosome is not anymore discuss in the document, I suggest to remove their description. It is adding confusion. In part 3.1, I don’t understand this part? Morevover, the “seed and soil” theory is also discussed in part 3.5. I suggest to remove 3.1. Part 3.2 do not talk about “pre-metastatic niche” but about cargo and surface protein. This part should be moved in the description part of the exosome (part 2) and some sentences moved in part 3.5 about metastasis. In part 4, it is confusing to regroup boron neutron capture therapy and immunotherapy. I suggest to split this part in 2. In part 4.3, I do not get the point about miR-134. Why the authors talk here about that? There also some duplication/repetition in this part. In part 4.4, it is not clear if SCDSFs are included into exosomes. If yes please clarify, if not please remove this part.Author Response
Answers to reviewer 4
Comment
The review proposed by Nicolini et al. was improved. However, there are still some problems with several parts and most of the titles does not reflect their content. In part 3, since oncosome is not anymore discuss in the document, I suggest to remove their description. It is adding confusion.
Answer
In part 3 the description of “oncosomes” has been removed. See section 3, page 6.
Comment
In part 3.1, I don't understand this part? Moreover, the "seed and soil" theory is also discussed in part 3.5. I suggest to remove 3.1.
Answer
Part 3.1. has been removed
Comment
Part 3.2 do not talk about "pre-metastatic niche" but about cargo and surface protein. This part should be moved in the description part of the exosome (part 2) and some sentences moved in part 3.5 about metastasis.
Answer
Part 3.2. has been completely removed as suggested. Particularly, the following sentences “Exosomes impede enzymatic degradation of their cargo by moving into the blood or extracellular environment [66] while p53 has been reported to have a relevant, although unknown, role in exosome secretion [84]. Neutral sphingomyelinase 2 (nSMase2) is an enzyme limiting ceramide biosynthesis, and its inhibition by ceramides affects exosomal release [85-87]” have been moved to take part of section 2.1. entitled “The biogenesis of exosomes”. See section 2.1, pages 3-4.
The paragraph “Cancer cells over-express EV-associated biogenesis machinery, which allows the release of more exosomes than from normal cells. ESCRT,YKT6,amplifying Rho/EGFRvIII,syntenin,heparanase,H-RASV12 and proto-oncogene Src signalling are constituents of this machinery [40,71-79]. Hypoxia and a low tumor-microenvironmental pH have also been found to favour exosomal traffic, particularly exosomal delivery and uptake in tumour cells [80-83].” has been moved to take part of section 3.2. entitled “Exosomes in cancer development and progression”. See section 3.2, page 8.
The paragraph “It has been described that tumour-derived exosomes favour the establishment of pre-metastatic niches by inducing the activation of Src, and up-regulation of pro-inflammatory S100 genes in the involved cells of target organs. Similarly, TGF hyper-expression of Kupffer cells in the liver by MIF-1 carried out by pancreatic tumour exosomes has been found. Subsequently, it has been observed that stellate cells produce an enhanced amount of fibronectin [57]. The remodelled TME further increases the arrival of macrophages from bone marrow, thus favouring the arising of a pre-metastatic niche in the liver [57].” have been moved to take part of section 3.3. entitled “Exosomes and cancer metastasis”. See page 10. The remaining part of this section has been moved to take part of section 2.2. entitled “Main characteristics of exosomes”. See pages 4-5.
Comment
In part 4, it is confusing to regroup boron neutron capture therapy and immunotherapy. I suggest to split this part in 2.
Answer
Part 4. has been splitted as suggested. The former became section 4.2. entitled “Exosomes as carriers of boron neutron capture therapy (BNCT) and the latter section 4.3. entitled “E xosomes as anticancer vaccines for immunotherapy”.
Comment
In part 4.3, I do not get the point about miR-134. Why the authors talk here about that? There also some duplication/repetition in this part.
Answer
The point about miR-134 has been removed. In particular the following paragraph “A report in 2015 tested the properties of miR-134 derived from exosomes of patients with triple-negative breast cancer (TNBC). The study detailed the therapeutic potential that miR-134 has as tumour suppressor and confirmed that miR-134 reduced TNBC aggressiveness and increased drug sensitivity [174]. Functional studies indicated that miR-134 delivered by direct transfection into Hs578Ts(i)8 cells reduced STAT5B, Hsp90 and Bcl-2 levels, counteracting cellular proliferation and enhancing cisplatin-induced apoptosis” has been removed. See page 15.
Comment
In part 4.4, it is not clear if SCDSFs are included into exosomes. If yes please clarify, if not please remove this part.
Answer
Regarding the SCDSFs and their role in reprogramming cancer cells we have described the results obtained in this field of research because the results which we have obtained are very important in increasing the possibility in ameliorating cancer treatments. We are now conducting different experiments using SCDSFs as cargoes of exosomes and we are studying the possibility to further increase the results already obtained. This is clearly described in the final part of the article in which we have recorded the researches which are now in progress. To clarify in better way this concept we propose to edit this part of the text. Instead of:"In this perspective, we put forward the proposal to use SCDSFs by loading them into exosomes to verify if their therapeutic effects could be increased. These studies are ongoing, and we will verify in the near future whether we will be able to obtain even more significant results at the level of biological cancer therapies by loading SCDSFs into exosomes” we would like to modify the sentence above as: We are now conducting experiments introducing SCDFs into exosomes and we are verifying if we can obtain better results both at experimental and clinical level. These researches are ongoing and we will illustrate the results obtained from these researches as soon as they are completed. See section 4.7, page 18.
Round 2
Reviewer 4 Report
The authors responded to most of my comments. However, I still believe that the last part (4.5) do not concern exosomes. Since authors wand to keep it and I understand the novelty of this part and its interest, I suggest to modify the 4.5 title with something like : "Stem cell differentiation stage factors (SCDSFs): others reprogramming factors of cancers, beside exosomes"
Author Response
As requested by the Reviewer, the title of section 4.5. "Stem Cells Differentiation Stage Factors (SCDSFs) and exosomes: their role in reprogramming cancer cells" has been replaced as suggested with "Stem Cells Differentiation Stage Factors (SCDSFs): other reprogramming factors of cancers, besides exosomes".This manuscript is a resubmission of an earlier submission. The following is a list of the peer review reports and author responses from that submission.
Round 1
Reviewer 1 Report
In this interesting review, Nicolini A. and co-authors describe the most crucial scientific community results obtained in these last years regarding the exosome role in solid tumors. The manuscript analyzes a lot of aspects, in particular, the role of exosomes in TME modulation and the potential of exosomes as carriers of stem cell differentiation. Only a few comments:
- Some aspects should be described/discuss in more details:
- A) Comparison of tissue profiles and tumor exosome profiles; I am referring mainly to smallRNA sequencing data. There are divergent "opinions": some researchers state that exosomes represent a picture of tumor tissue and others do not.
B) Patients ' tumor-derived exosomes: the authors should comment on the origin of tumor exosomes. In the vast majority of published papers, it is impossible to establish exosomes' source; are exosomes from cancer patients or exosomes produced by the tumor ?
- page 3, line 94: it was demonstrated that exosome cargo is represented not only by miRNA, but also by other small RNA subtypes, such as YRNA , tRNA ....please modify the content between brackets and relative references
Reviewer 2 Report
Exosomes and Cancer: bio-informational reprogramming of malignant cells, a new pathway for solid tumors therapy? A. Nicolini et al. Cancers. Ref. Manuscript ID: cancers-975262.
The authors have put together a reviewed the literature on Exosome and cancer highlighting the importance of malignant cell reprogramming and its implication for solid tumors therapy. This article represents a comprehensive review. There are minor points which should be addressed:
- Text in Figure 1 is not legible; the figure overall should be improvised
- The article should undergo thorough editing particularly for ease of understanding – several places it’s hard to understand: For example,
- Lines 152-154 – “Nucleic acids in exosomes……………………………..”
- Line 546: “Cancer cells seem can convert…………………………………..”
Reviewer 3 Report
The manuscript Exosomes and cancer: bio-informational reprogramming of malignant cells, a new pathway for solid tumors therapy? by Andrea Nicolini et al. addresses the role of exosomes in carcinogenesis and the use of exosomes in cancer therapy. The topic is novel and highly interesting for the readers of Cancers journal. I am not native English speaker, however, the English language in some sections deserves attention and some reorganization of the order the following chapters could perhaps help the reader to better catch the key messages.
1.Title
The title of the manuscript specifically mentions therapy of solid tumours only. Why? Conclusions should give answer to this. If not, solid tumours should not be mentioned in the title. Title as a whole takes up the reprogramming of malignant cells and this choice should be more visiblyjustified in the text.
2.Abstract(s), Introduction
The English grammar and style as should be checked and some typos corrected in the abstract, introduction and conclusions. These sections are written in a different style compared to Chapters 2, 3 and 4 which read really well. The abstract, introduction and conclusions are very important elements of a Review that should raise the interest of the reader to learn more on the content. Currently, the text needs clarity to highlight -and indeed make justice for -the actual scientific content of the manuscript which is very timely and excellent.Some suggestions for such editorial changes are below.
3.Chapters 2-4
Chapter 2 gives an introduction to the biogenesis and principal characteristics of exosomes. This chapter is clearly written and sets the scene for the next chapters. Figure 1 is informative however the font may be too small to show up in the final edition.
Chapter 3 (Tumour-derived exosomes (oncosomes)) tells about the role of exosomes in carcinogenesis. “The role of exosomes in carcinogenesis” could be considered as alternative heading for Chapter 3, as the main content is about their function rather than their characteristics which are shortly covered in the very beginning of the chapter.Table 1 is very informative.
The title of Chapter 4 “Exosomes and reprogramming treatments in cancer treatments” seems too narrow compared to the actual content. Perhaps something like “Use of exosomes in cancer therapy” fits better, where 4.1 already gives an overall introduction to the different options and Table 2 lists experimental studies and clinical trials for exosomal cancer therapy. The very high specificity in targeting cancer cells could be further emphasized. Here some reference could be made to radiation therapies like targeted BNCT where boron can be carried to target cells by exosomes. Immune enhancer therapy described in 4.2 is different from the cancer cell reprogramming approaches described in chapters 4.4 and 4.5 which are based on epigenetic regulation forcing the cancer cells to differentiate. No cancers arise from differentiated cells.
4.Conclusions
The conclusions would benefit of structured discussion. For example, the specificity of the use of exosomes in cancer treatment should be discussed both in terms of finding the specific target cells and the specific treatment for the particular cancer types.Also,it should be clear when the exosome discussion is about the biological properties and when about the application of exosomes in therapy. Statement “In addition, exosomes could acquire important factors from different types of stem cells and cause differentiation or apoptosis in cancer cells. These factors are the most important part, since they are the ones that can cause beneficial phenotypic modifications to the cells” seems premature, given the fact that the clinical trials with exosomes listed in Table 2 are on cytotoxic effects and immune responses, not differentiation of cancer cells. Conclusions should follow from the evidence presented in the previous chapters. Like Abstracts and the introduction, this section also needs language check.
Reviewer 4 Report
Nicoli et al. wrote a review about exosome and cancer. There are so many reviews about this topic but the originality of their review could be based on the discussion about the reprogramming of the cancer cells announced in the title. However, this aspect is discussed only on the last paragraph and does not speak specifically about exosomes, or extracellular vesicles, but about proteins from stem cells. This is very confusing.
Overall, the plan is difficult to follow with many unorganized parts talking about some cells, then others and next go back to the first ones.
The authors don’t follow MISEV guidelines and sometimes called exosomes, some large vesicles or oncosomes, which nevertheless all meet specific criteria.
Some parts are repeated, others for example in the immune system paragraph don’t talk at all about immunity, most of the sentences in the treatment part do not describe some treatments and finally recent publications in a topic which evolves quickly are not cited and discussed.